# Relationship between Bacterial Vaginosis and Sexually Transmitted Infections: Coincidence, Consequence or Co-Transmission?

**DOI:** 10.3390/microorganisms11102470

**Published:** 2023-10-01

**Authors:** Linda Abou Chacra, Claudia Ly, Alissa Hammoud, Rim Iwaza, Oleg Mediannikov, Florence Bretelle, Florence Fenollar

**Affiliations:** 1Campus Santé Timone, Aix-Marseille University, IRD, AP-HM, SSA, VITROME, 13005 Marseille, France; abouchacra.linda@gmail.com (L.A.C.); claudiaks_ly@hotmail.com (C.L.); 2IHU-Méditerranée Infection, 13005 Marseille, Francereem.iwaza@gmail.com (R.I.); olegusss1@gmail.com (O.M.); 3Campus Santé Timone, Aix-Marseille University, IRD, AP-HM, MEPHI, 13005 Marseille, France; florence.bretelle@ap-hm.fr; 4Department of Gynaecology and Obstetrics, Gynépole, La Conception, AP-HM, 13005 Marseille, France

**Keywords:** vaginal ecosystem, bacterial vaginosis, dysbiosis, sexually transmitted infection, sexual practices

## Abstract

Sexually transmitted infections (STIs) are a serious global problem, causing disease, suffering, and death. Although bacterial vaginosis (BV) is not considered to be an STI, it may be associated with an increased risk of contracting a wide range of STIs. We sought to assess the link between the different microorganisms involved in STIs and BV. A total of 290 vaginal swabs from 290 women sent for diagnostic purposes to the clinical microbiology laboratory of the Marseille University Public Hospitals were tested by specific qPCR targeting STI-causing microorganisms and BV. Of these 290 swabs, 15.2% (44/290) were diagnosed with at least one STI-causing microorganism and 17.2% (50/290) with BV. The prevalence of STIs was significantly higher in women with BV (28%, 14/50) than in those without (20.4%, 51/240). The prevalence of co-infections involving two STI-causing microorganisms was significantly more frequent in women with BV than in those without (18% [8/50] vs. 2% [5/250]; *p* < 0.001). The prevalence of monoinfections and polyinfections with STI-causing microorganisms was lower in women without BV than in those with (8.8% [21/240] vs. 28% [14/50]), *p* < 0.001 and 2% (5/240) vs. 8% (4/50), *p* = 0.05, respectively). Our data suggest that a correlation between BV and STI may exist, with a higher prevalence of both monoinfections and polyinfections involving STI-causing microorganisms in women with BV. Further research is needed to better understand BV and its links to STIs.

## 1. Introduction

It has long been known that the vaginal microflora plays a crucial role in maintaining a normal physiological environment for the host and that its participation is considered essential for successful reproduction [1]. In healthy women of childbearing age, the normal vaginal flora is classically dominated by *lactobacilli*. Between 1 and 4 mL of vaginal fluid can contain 10^8^ to 10^9^ bacterial cells, which play a key role in maintaining the physiological balance of the female genital tract and protecting the vaginal mucosa. Indeed, *lactobacilli*, with their production of antimicrobials, as well as those of lactic acid from the glycogen contained in the vaginal mucosa, making it possible to generate an acid vaginal pH (of 4.5 or less), provide an effective first-line defence against potentially pathogenic microorganisms [1]. Currently, the complex interaction between vaginal microflora and reproductive health is a major research topic in the fields of medicine and biology and continues to generate considerable interest in better understanding its impact on women’s health.

Bacterial vaginosis is a disturbance of the vaginal ecosystem, characterised by a significant depletion of the *lactobacilli* due to a strong proliferation of anaerobic bacteria, which are less abundant in the normal vaginal flora, such as *Gardnerella vaginalis*, *Fannyhessea* (formerly *Atopobium*) *vaginae*, *Ureaplasma urealyticum*, *Mycoplasma hominis*, *Prevotella*, *Peptoniphilus*, *Megasphaera*, *Mobiluncus*, and several fastidious bacteria characterised as “Bacterial Vaginosis Associated Bacteria” (BVAB-1, BVAB-2, and BVAB-3) [2]. The exact aetiology of these changes has remained elusive to date, although many trigger factors are related, such as sexual activities [3], vaginal cleansing [4], contraception [5] and antibiotic use [6]. The prevalence of bacterial vaginosis varies between countries around the world, ranging from around 10% to 30% among women who have sex with men (WSM) [7,8] and 25% to 50% among women who have sex with women (WSW) [9], with the significant majority of women with bacterial vaginosis being asymptomatic [10]. The highest prevalence is reported in sub-Saharan Africa [11]. In Europe, the prevalence of bacterial vaginosis is much lower [12]. Bacterial vaginosis occurs frequently in women of childbearing age [13].

The rise in the incidence of sexually transmitted infections by more than one million new cases every day [14] has led to an alarming major public health crisis. This increase in sexually transmitted infections is a cause for concern, as sexual contact can spread more than 30 different bacteria, viruses, and parasites [15]. There is a direct impact on sexual and reproductive health caused by sexually transmitted infections in women’s upper and lower reproductive tracts [15]. Asymptomatic sexually transmitted infections are a cause for concern, as they are difficult to identify [15]. Their non-negligible prevalence is also a worrying aspect. Indeed, these particularly insidious infections cannot be identified without specific tests and can therefore easily spread in a sexually active population [15].

Although bacterial vaginosis is not considered to be a sexually transmitted infection, this dysbiotic condition may be associated with an increased risk of contracting a wide range of sexually transmitted infections [16,17]. For example, a cohort study of 255 non-pregnant women showed that bacterial vaginosis was associated with a 3.4- and 4-fold increased risk of positive tests for *Chlamydia trachomatis* and *Neisseria gonorrhoeae*, respectively [18]. Similarly, Brotman et al. demonstrated that women with bacterial-vaginosis-associated vaginal microbiota or low levels of *lactobacilli* had the highest relative proportion of human-papillomavirus-positive samples [19].

In this paper, we sought to study the concomitance of bacterial vaginosis and sexually transmitted infections in women and compared samples from women with bacterial vaginosis versus women without, using molecular biology as a rational diagnostic tool.

## 2. Materials and Methods

### 2.1. Data Collection and Ethical Considerations

A total of 290 vaginal specimens were sampled from 290 women using Sigma Transwab^®^ swabs (Medical Wire, Corsham, UK). These specimens were initially collected for diagnostic purposes and sent to the clinical microbiology laboratory at our University Public Hospitals (Assistance Publique—Hôpitaux de Marseille [AP-HM], Marseille, France). The data analysed in this study are thus retrospective and cover a six-month period in 2022. Before testing, these samples were meticulously stored at −80 °C to preserve their integrity.

This is, therefore, a retrospective study of samples initially obtained during patient management, in accordance with Article L.1211-2 of the French Public Health Code. Patients were informed of the potential secondary use of their samples and personal data, collected as part of their clinical care, for research purposes. They also had the possibility of refusing this secondary use by notifying the AP-HM data protection officer of their objections. In addition, all personal data relating to patients have been anonymized to protect their privacy and their confidentiality. The study was conducted in accordance with the Declaration of Helsinki. The entire protocol has been approved by our independent ethics committee (IEC no. 2022-033).

### 2.2. Molecular Strategies for Detecting Microorganisms

The molecular strategy used in this study to detect microorganisms involved a complete set of target genes, each with its corresponding probes and primers. Eleven DNA sequences targeting ten microorganisms and one human gene were analysed using real-time quantitative PCR assay. Quantification of the human albumin gene with exon 12 as the target sequence was thus used as an internal control to provide evidence of DNA presence and quality, as well as the potential presence of PCR inhibitors. The microorganisms tested, with their target sequences in brackets, were as follows: *G. vaginalis* (*Cpn* 60), *F. vaginae* (16S rRNA), *Lactobacillus* spp. (*Tuf*), *Mycoplasma genitalium* (*FusA*), *Trichomonas vaginalis* (repeated sequence), *N. gonorrhoeae* (hypothetical protein), C. *trachomatis* (hypothetical protein), *Treponema pallidum* (*PolA*), *Haemophilus ducreyi* (*GroESL*), and Herpes simplex virus (*Pol*). The list of these target sequences and their respective probes and primers sequences used are more detailed in the Appendix A. To ensure accuracy and reliability of results, synthetic positive controls (plasmids) and negative controls (master mixes) were also included in each PCR run. Data were systematically interpreted after validation of the quality of these positive and negative controls. This rigorous quality control strategy ensures the credibility of the data generated by the study.

DNA extraction from vaginal samples was performed using a Qiagen EZ1 bio-robot (Qiagen, Courtaboeuf, France) in conjunction with a commercial extraction kit known as the QIAamp Tissue Kit^®^ (Qiagen), following the precise guidelines defined by the kit’s manufacturer. The DNA extraction process began with the digestion of 100 µL of vaginal samples using 100 μL of G2 buffer and 10 μL of proteinase K at a controlled temperature of 56 °C for a period of 20 min. Next, the elution step yielded 100 µL of DNA extracts, which were then carefully stored at −20 °C to preserve their stability and integrity until subsequent analysis.

Specific quantitative real-time polymerase chain reaction (PCR) was performed using a CFX96 Real-Time system (Bio-Rad Laboratories, Foster City, CA, USA). DNA amplification was performed based on the following thermal profile: (1) incubation was carried out at 50 °C for two minutes (for Uracil DNA Glycosylase (UDG) activation); (2) initial DNA denaturation was performed at 95 °C for five minutes; and, (3) finally, a series of 39 cycles consisting of DNA denaturation at 95 °C for five seconds and primer annealing-probe hybridization at 60 °C for 30 s was carried out. The final reaction mix volume for quantitative real-time PCR assays contained 10 μL of Eurogentec™ Probe PCR Master Mix (Eurogentec, Liege, Belgium), 3 μL of distilled water DNAse and RNAse free, 0.5 μL of each reverse and forward primers (50 μM), probe (50 μM), UDG, and 5 µL of extracted DNA.

For *C. trachomatis*, *M. genitalium*, *N. gonorrhoeae*, *T. vaginalis*, *H. ducreyi*, and *T. pallidum*, as well as for HSV, samples were considered positive for the targeted microorganisms if the cycle threshold (ct) value was lower than 35. In order to detect the presence of bacterial vaginosis, a quantitative real-time PCR was applied to all DNA vaginal samples to quantify both *F. vaginae* [20] and *G. vaginalis* DNA levels, as previously reported [21,22]. As mentioned in the initial articles, results were expressed as copies of microbial DNA per 1 mL of vaginal suspension. In addition, the synthetic positive control (dilution to 10^4^ of a quantification plasmid) was used for quantification. Finally, according to the strategy presented by Ménard et al., the diagnosis of bacterial vaginosis flora was made when bacterial loads ≥ 10^8^ DNA copies/mL for *F. vaginae* and/or ≥10^9^ DNA copies/mL for *G. vaginalis* were detected in vaginal samples. This patented strategy has been routinely used as the gold standard diagnostic tool for bacterial vaginosis in our clinical microbiology laboratory for many years.

### 2.3. Statistical Analysis

IBM SPSS Statistics version 23 served as the primary tool for statistical analysis. The Chi-square test was employed to assess the potential associations between bacterial vaginosis and sexually transmitted infections among the study participants. In cases in which the effective sample sizes were low, Fischer’s exact test was applied to maintain statistical accuracy. All analyses were conducted using bilateral *p*-values, with statistical significance set at *p* ≤ 0.05 to ensure the robustness and reliability of our findings.

## 3. Results

### 3.1. Main Patient Characteristics

Among the 290 women, 222 (76.6%) were pregnant and 68 (23.4%) were women of childbearing age with no pregnancies to date. Aged varied from 18 to 49 years old (median age: 28 years old, mean age: 28.7 years old). Most of the women were asymptomatic 243/290 (83.7%).

### 3.2. Molecular Results

Bacterial vaginosis flora was diagnosed in 50 (17.2%) of the 290 women (Table 1). Forty-five of them (90%) had bacterial loads of at least 10^8^ and 10^9^ DNA copies/mL for *F. vaginae* and *G. vaginalis*, respectively. Three (6%) had a bacterial load of at least 10^8^ DNA copies/mL for *F. vaginae* and two (4%) of at least 10^9^ DNA copies/mL for *G. vaginalis*. Thirty-six women (72%) of the fifty with bacterial vaginosis flora had no complaints of vaginal discharge. The patients’ ages varied between 18 and 48 years old. No age difference was observed between women with and without bacterial vaginosis flora (28.9 ± 6.9 vs. 30.2 ± 9.5, respectively; *p* = 0.35).

A total of 44 (15.2%) of the 290 women were diagnosed with at least one microorganism involved in sexually transmitted infections. Thirty-six of them (81.8%) were asymptomatic. Overall, the prevalence was 6.5% (19/290) for *C. trachomatis*, 1.4% (4/290) for *N. gonorrhoeae*, 7.9% (23/290) for *T. vaginalis*, and 2.4% (7/290) for *M. genitalium* (Table 2).

The majority (35/44 [79.5%]) of patients were infected by only one sexually transmitted infection and only 9/44 (20.5%) were also co-infected by several pathogens.

### 3.3. Bacterial Vaginosis and Sexually Transmitted Microorganisms

Bacterial vaginosis flora was significantly associated with microorganisms involved in sexually transmitted infections (Table 3). Indeed, the prevalence of sexually transmitted infections in women with bacterial vaginosis flora was 36% (18/50) vs. 10.8% (26/240) in women without bacterial vaginosis flora (*p* < 0.001). *C. trachomatis* and *N. gonorrhoeae* were significantly more frequently detected in the 50 women with bacterial vaginosis flora (24% [12 positive]; 6% [3]) than the 240 without (2.9% [7]; 0.4% [1]), *p* < 0.001, *p* = 0.01, respectively. Although *T. vaginalis* and *M. genitalium* were more frequently detected in women with bacterial vaginosis flora (10% [5]; 4% [2]) than in those without (7.5% [18]; 2.1% [5]), this was not significant (*p* = 0.34, *p* = 0.56, respectively).

The distribution of co-infections due to sexually transmitted infection-causing microorganisms was also substantially related to the state of the vaginal flora. In women without bacterial vaginosis flora, the prevalence of monoinfections with sexually transmitted infection-causing microorganisms was 8.8% (21/240) compared to 28% (14/50) in women with bacterial vaginosis flora (*p* < 0.001). Meanwhile, the prevalence of polyinfections was 2% (5/240) in women without bacterial vaginosis flora versus 8% (4/50) in women with bacterial vaginosis flora (*p* = 0.05) (Table 3).

## 4. Discussion

All our data are based on rational molecular methods to evaluate the prevalence of bacterial vaginosis, as well as to study the association between this dysbiosis with four microorganisms involved in sexually transmitted infections. Until now, studies on the association between bacterial vaginosis and sexually transmitted infections have traditionally used the Nugent score or the Amsel criteria for the diagnosis of bacterial vaginosis flora [17,23,24].

However, a certain number of shortcomings of these strategies (subjective interpretation influenced by individual skills and ambiguity in the classification of the intermediate flora, etc.) led to the conclusion that these strategies were outdated [25,26]. Molecular biology is therefore increasingly practiced for the diagnosis of bacterial vaginosis flora because it is objective and rational while being able to be carried out in point-of-care laboratories [25,26].

In our series, the prevalence of bacterial vaginosis and sexually transmitted infections was 17.2% and 15.2%, respectively. The prevalence of bacterial vaginosis in our study in France is similar to that which has been observed in Western Europe, where it is generally below 20% [12]. For sexually transmitted infections, available data indicate that they are again on the rise in many EU countries and have been since the mid-1990s [27,28]. Compared with 2015, reported cases of sexually transmitted infections rose significantly in 2019. Data show an increase of 9% for *Chlamydia* infections, 55% for gonorrhoea, 25% for syphilis and 75% for lymphogranuloma venereum [28]. None of our patients had genital ulcerations, and although we looked for *H. ducreyi*, *T. pallidum* and HSV out of curiosity, as a matter of routine, no cases were detected. This is consistent with current data showing that syndromic management has reduced the prevalence of sexually transmitted microorganisms associated with genital ulcers [29]. It should be emphasised that most women diagnosed with bacterial vaginosis flora and sexually transmitted infections in our series were asymptomatic (72% and 81.5%, respectively), including those with associated polyinfections (Table 4). Moreover, the extent of clinical manifestations does not appear to be directly proportional to the number of pathogens present.

This highlights two main points:(1)The difficulties in preventing the circulation of microorganisms implicated in sexually transmitted infections not associated with genital ulcers [29]. Indeed, sexually transmitted infections not associated with genital ulcers present a major challenge in terms of prevention. Unlike sexually transmitted infections with visible symptoms such as genital ulcers, asymptomatic or minimally symptomatic sexually transmitted infections can spread silently, making them difficult to detect and control. This raises questions on the strategies prevention existing and suggests the necessity prevention approaches more targeted and innovative, as well as an increased awareness to encourage regular screening, in particular in high-risk populations.(2)The difficulty of understanding the impact of bacterial vaginosis in obstetric complications such as prematurity [30,31,32]. The relationship between bacterial vaginosis and obstetric complications, particularly prematurity, remains complex and incompletely understood. Bacterial vaginosis can lead to imbalances in vaginal microflora, but how this contributes to adverse obstetric outcomes requires further research. Understanding these mechanisms is essential for developing effective preventive interventions and care protocols for pregnant women with bacterial vaginosis, with the aim of reducing the risk of prematurity and other associated complications.

Previous studies generally acknowledge that there is an increased prevalence of sexually transmitted infection-causing microorganisms in women with bacterial vaginosis [33,34,35,36,37,38,39,40]. Our data show that women with bacterial vaginosis were 1.79 times more likely to be positive for sexually transmitted infections. Overall, *C. trachomatis* and *N. gonorrheae* have a significantly higher prevalence in women with bacterial vaginosis than in those without bacterial vaginosis, while *T. vaginalis* and *M. genitalium* were positively but not significantly correlated with bacterial vaginosis. In their study of 95 sexually transmitted infection-positive and 91 sexually transmitted infection-negative samples, Shipitsyna et al. showed a significant age-independent association between bacterial vaginosis-associated vaginal microbiota and the presence of *C. trachomatis*, *M. genitalium*, and *T. vaginalis* [33]. In 2021, in a systematic review of 14 articles and a meta-analysis, Seña et al. demonstrated that the risk of *T. vaginalis* is almost twice as high in women with bacterial vaginosis than in women without bacterial vaginosis; however, studies note heterogeneity and potential confounding factors (age, sexual partners, etc.) [39]. Nye et al. showed that the prevalence of *M. genitalium* infection in patients with bacterial vaginosis was significantly higher than in those without bacterial vaginosis (7.0% [20/307] vs. 3.6% [42/1225]) [38].

A total of 3.2% of the women exhibited multiple infections with sexually transmitted infection-causing microorganisms. Among sexually transmitted infections, the co-infection with *C. trachomatis* was the most frequent. In women with bacterial vaginosis, co-infections have a much higher prevalence than in women without. Sexually transmitted infection-causing microorganisms commonly circulate. A much higher frequency is also observed in women with bacterial vaginosis. Although most studies suggest bacterial vaginosis as a risk factor for sexually transmitted infections, linked to local inflammation, the question that arises is whether bacterial vaginosis is a cause, a consequence, or constitutes a sexually transmitted infection in its own right. Although the lack of a known causative agent makes it difficult to classify bacterial vaginosis as a sexually transmitted infection, it is strongly linked to sexual behaviours and exhibits some of the hallmarks of a sexually transmitted infection [40].

As early as 1955, when Gardner and Dukes conducted their first exhaustive study of bacterial vaginosis, they made some intriguing findings. Indeed, they reported the possibility of women with normal vaginal flora being infected with material from the vaginas of patients with bacterial vaginosis. This observation raised vital questions about the transmission of this condition and its potential link with sexual behaviour. In addition, Gardner and Dukes also highlighted the protective effect of condom use, suggesting that sexual practices may play a key role in the transmission of bacterial vaginosis. Another striking feature of their findings was the development of bacterial vaginosis in patients shortly after marriage, whereas their vaginal flora was normal at the time of the premarital examination [41]. Since then, most studies focusing on the risk factors for bacterial vaginosis have converged on an intriguing conclusion: the risk profile of this condition seems to strikingly mirror that of sexually transmitted infections, further underlining the complex links between bacterial vaginosis and female reproductive health. These findings are summarized in Appendix A [7,40,41,42,43,44,45,46,47,48,49,50,51,52,53,54,55,56,57,58,59,60,61,62,63,64,65,66,67,68,69,70,71,72,73,74,75,76,77,78], underlining the continuing need for in-depth research to elucidate the nature of this complex association.

One of the limitations of our study is the lack of data on participants’ behavioural characteristics, including their sexual practices, to analyse their potential influence. In addition, no data about partners are available, nor are follow-up data available. There is, therefore, only one time point for evaluation. In conclusion, there is a significant correlation between bacterial vaginosis flora and sexually transmitted infections. The majority of women remain asymptomatic, including those with multiple infections. Multiple sexually transmitted infections are significantly more frequent in patients with bacterial vaginosis flora than in those with normal vaginal flora. This fact sheds light on the presentation of bacterial vaginosis as a predisposing factor for sexually transmitted infections and one that is linked with inflammation may be too mechanistic. This study found the association of bacterial vaginosis with isolated monoinfections or multiple infections, aligning bacterial vaginosis with transmitted infectious diseases.

Although no specific bacteria can be directly associated with bacterial vaginosis, and bacterial vaginosis itself cannot be categorically classified as a sexually transmitted infection, there is a clear link between this condition and sexual intercourse. At present, our understanding of the mechanisms underlying this complex relationship remains limited. It is therefore imperative to pursue in-depth research to unravel this enigma and clarify the precise links between bacterial vaginosis and sexually transmitted infections. This increased knowledge is essential to guide prevention and treatment strategies, and to improve women’s sexual health.

## Figures and Tables

**Table 1 microorganisms-11-02470-t001:** Prevalence of bacterial vaginosis flora among 290 women according to their status.

	Number of Positive/Total Number Tested	Reproductive Age without Pregnancy	Pregnancy
Bacterial Vaginosis Flora	50/290 (17.2%)	11/68 (16.2%)	39/222 (17.6%)

**Table 2 microorganisms-11-02470-t002:** Representation of the microorganism’s prevalence that are involved in sexually transmitted infections, in normal flora and bacterial vaginosis flora, in pregnant and non-pregnant women.

	Number of Positive Samples (Percentage)			
	Among All Vaginal Specimens Tested	According to Vaginal Flora Type		Fisher Test
Microorganisms	Total of 290 Women	240 Normal Flora	50 Bacterial Vaginosis Flora	*p*-value
*C. trachomatis*	19 (6.6%)	7 (2.9%)	12 (24%)	**<0.001 ***
*N. gonorrhoeae*	4 (1.4%)	1 (0.4%)	3 (6%)	**0.01 ***
*M. genitalium*	7 (2.4%)	5 (2.1%)	2 (4%)	0.34
*T. vaginalis*	23 (7.9%)	18 (7.5%)	5 (10%)	0.56
	222 Pregnant Women	183 Normal Flora	39 Bacterial Vaginosis Flora	*p*-value
*C. trachomatis*	16 (7.2%)	6 (3.3%)	10 (25.6%)	**<0.001 ***
*N. gonorrhoeae*	0	0	0	0
*M. genitalium*	7 (3.2%)	5 (2.7%)	2 (5.1%)	0.35
*T. vaginalis*	16 (7.2%)	12 (6.6%)	4 (10.3%)	0.49
	68 Non-Pregnant Women	57 Normal Flora	11 Bacterial Vaginosis Flora	*p*-value
*C. trachomatis*	3 (4.4%)	1 (1.8%)	2 (18.2%)	0.06
*N. gonorrhoeae*	4 (5.9%)	1 (1.8%)	3 (27.3%)	**0.01 ***
*M. genitalium*	0	0	0	0
*T. vaginalis*	7 (10.3%)	6 (10.5%)	1 (9.1%)	1

* Significant *p*-value in bold (*p* < 0.05).

**Table 3 microorganisms-11-02470-t003:** Prevalence of microorganisms involved in sexually transmitted infections, either as monomicrobial or polymicrobial infections, in pregnant women and in non-pregnant women. Data presented for normal flora and bacterial vaginosis flora.

	**Number of Positive Samples (Percentage)**			
	**222 Pregnant Women**	**183 Normal Flora**	**39 Bacterial Vaginosis Flora**	** *p* ** **-Value**
Microorganisms involved in sexually transmitted infections	25 (11.3%)	15 (8.2%)	10 (25.6%)	-
*T. vaginalis*	10 (4.5%)	9 (4.9%)	1 (2.6%)	-
*C. trachomatis*	12 (5.4%)	4 (2.2%)	8 (20.5%)	-
*N. gonorrhoeae*	1 (0.4%)	0	1 (2.6%)	-
*M. genitalium*	2 (0.9%)	2 (1.1%)	0	-
2 microorganisms involved in sexually transmitted infections	7 (3.1%)	4 (2.2%)	3 (7.7%)	**<0.001**
*T. vaginalis*/*M. genitalium*	3 (1.3%)	2 (1.1%)	1 (2.6%)	-
*C. trachomatis*/*T. vaginalis*	3 (1.3%)	1 (0.6%)	2 (5.1%)	-
*C. trachomatis*/*M. genitalium*	1 (0.4%)	1 (0.6%)	0	-
*C. trachomatis*/*N. gonorrhoeae*	0	0	0	-
Total	32 (14.4%)	19 (10.4%)	13 (33.3%)	**<0.001**
	**68 Non-Pregnant Women**	**57 Normal Flora**	**11 Bacterial Vaginosis Flora**	** *p* ** **-Value**
Microorganisms involved in sexually transmitted infections	10 (14.7%)	6 (10.5%)	4 (36.4%)	-
*T. vaginalis*	6 (8.8%)	6 (10.5%)	0	-
*C. trachomatis*	1 (1.5%)	0	1 (9.1%)	-
*N. gonorrhoeae*	3 (4.4%)	0	3 (27.3%)	-
*M. genitalium*	0	0	0	-
2 microorganisms involved in sexually transmitted infections	2 (2.9%)	1 (1.7%)	1 (9.1%)	**<0.001**
*T. vaginalis*/*M. genitalium*	0	0	0	-
*C. trachomatis*/*T. vaginalis*	1 (1.5%)	0	1 (9.1%)	-
*C. trachomatis*/*M. genitalium*	0	0	0	-
*C. trachomatis*/*N. gonorrhoeae*	1 (1.5%)	1 (1.7%)	0	-
Total	12 (17.6%)	7 (12.2%)	5 (45.5%)	**<0.001**

**Table 4 microorganisms-11-02470-t004:** Distribution of asymptomatic women according to vaginal flora types and microorganisms involved in sexually transmitted infections.

	Number of Asymptomatic Women According to Vaginal Flora Type (Percentage)		
	Normal Flora	Bacterial Vaginosis Flora	Total
	207/240 (86.2%)	36/50 (72%)	243/290 (83.8%)
Free of microorganisms involved in sexually transmitted infections	182/214 (75.5%)	25/32 (78.1%)	153/246 (62.1%)
Monomicrobial infections	20/21 (95.2%)	9/14 (81.8%)	29/35 (82.8%)
*T. vaginalis*	15/15	0	15/16
*C. trachomatis*	3/4	7/9	10/13
*N. gonorrhoeae*	0	1/3	1/3
*M. genitalium*	2/2	1/1	3/3
Polymicrobial infections with 2 microorganisms	5/5 (100%)	2/4 (50%)	7/9 (77.8%)
*C. trachomatis/T. vaginalis*	1/1	2/3	3/4
*C. trachomatis/M. genitalium*	1/1	0	1/1
*C. trachomatis*/*N. gonorrhoeae*	1/1	0	1/1
*T. vaginalis*/*M. genitalium*	2/2	0	2/3

## Data Availability

All data is available on reasonable request from the corresponding author.

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
