# Peer review of "Relationship between Bacterial Vaginosis and Sexually Transmitted Infections: Coincidence, Consequence or Co-Transmission?"

_microorganisms, 2023, doi:10.3390/microorganisms11102470_

Round 1
Reviewer 1 Report
It is really hard to believe that the authors did not detect any G. vaginalis in the samples. I can only find some information in the material methods part.
The author should revise the manuscript extremely carefully.
Author Response
Pr Florence FENOLLAR
Aix-Marseille Université (AMU) - Faculté de Médecine, Marseille
Unité VITROME (Vecteurs – Infections Tropicales et Méditerranéennes)
IHU - Méditerranée Infection,
19-21 Boulevard Jean Moulin, 13005 Marseille
Email: florence.fenollar@univ-amu.fr
September 11th, 2023
Dear editor,
Thank you for giving us the opportunity to submit a revised version of our manuscript titled “
Relationship between bacterial vaginosis and sexually trans-mitted infections: coincidence, consequence or co-transmission?”. We are grateful for the time and effort that you and the reviewers have dedicated to providing your valuable feedback on our manuscript. We are indebted to the reviewers for their insightful comments on my review. We have been able to incorporate changes to reflect most of the suggestions provided by the reviewers. We have highlighted the changes within the manuscript. Here is a point-by-point response to the reviewers’ comments and concerns.
Comments for the author:
Reviewer #1:
It is really hard to believe that the authors did not detect any G. vaginalis in the samples. I can only find some information in the material methods part.
The author should revise the manuscript extremely carefully.
We appreciate the reviewer's thorough consideration of our manuscript. In our study, we used molecular diagnostic techniques targeting two specific microorganisms, G. vaginalis and F. vaginae, as indicators of BV. VB positivity was defined by the presence of these microorganisms.
We acknowledge that this information was mainly presented in the "materials and methods" section. Although we did not explicitly mention the detection of G. vaginalis in the results section, the presence or absence of G. vaginalis was indeed a crucial part of our analysis and contributed to the determination of BV status.
Sincerely yours,
Pr Florence FENOLLAR
Reviewer 2 Report
Authors intended to determine a relation between bacterial vaginosis and STI.
88 references for a paper that is not a revision does not make sense.
Abstract:
Authors should not state “There is a correlation”. Instead, they should state something like ‘Our data suggest that a correlation between BV and STI may exist’
“BV itself is not classified as an STI, it can be transmitted during sexual intercourse.” ??? If BV is, as authors state, “a disturbance of the vaginal ecosystem” how can a disturbance be transmitted? This sentence must disappear.
Introduction:
To consider BV as asymptomatic is incongruous. How can one put a name of a disease associated to a laboratory finding without any clinical complaints?
This is surely a problem when talking about BV. Not saying that Nugent or Amsel were better, but the use of molecular techniques to detect and quantify some bacteria (when, to date, the etiology of BV remains mostly unknown) is at least as misleading as older methods… considering some laboratory findings as BV (+) of women with no symptoms or clinical signs and women with symptoms of BV getting a BV (-) result…
Materials and Methods:
When were the vaginal samples taken? During how many days/weeks/months? How long were samples kept before testing? At what temperature?
Age of participants is wrongly in results. And it is repeated in 3.1 and 3.2
The age is too heterogeneous for this evaluation. Of course women of more than 70 years old or more might remain sexually active, but women of non-reproductive age (usually above 50) should be excluded, or their evaluation be separated from women 18 to 49. To mix pregnant and non-pregnant also seems incorrect.
Bacteria and Protozoan names should in italic in lines 105 and 106
To mix HPV among the STI agents is incorrect. Of course it is sexually transmitted but it is not part of the STI screening. How many years can HPV remain in the cervix?... It surely does not correlate with recent infection and this is a reason, among others, for being excluded from the routine STI screening. So, HPV should completely disappear from this paper. The same for Treponema pallidum, Haemophilus ducreyi and HSV1/2 because there is no sense for screening sexually transmitted etiological agents of genital ulcer... without the presence of… genital ulcers… So all the sentences regarding all these agents should disappear
M. hominis is part of the vaginal flora… So, why including it as “one microorganism” found? It is normal that it is there. Have authors quantified it?...
Ln 158 a number with two decimal places
Figure 1 is of null interest and when HPV disappears, it will no longer make sense
Discussion:
“between this dysbiosis with a large panel of microorganisms involved in STIs” large panel?... C. trachomatis, N. gonorrhoeae, T. vaginalis and M. genitalium are 4 and the only that make sense.
Authors cannot state “Our data are thus consistent.” ??? . Because authors positive and negative controls?... That is obligatory…
“For STIs, available data indicate that STIs are again on the rise in many EU countries and have been since the mid-1990s [29].” To use a paper published almost 20 years ago… in 2004… for stating that STI are on the rise in 2023… seems very bad. There are much more recent ‘available data’…
The sentence “Finally, one of the strong points of our study is the screening of many different pathogens based on rational molecular tools.” should disappear.
“there is a significant correlation between BV and STIs” when the study will be restricted to 4 STI agents (CT, NG , TV and MG), women 50 and over are discarded, and pregnant discarded or evaluated separately, authors will find out whether there is a correlation or not.
Author Response
Pr Florence FENOLLAR
Aix-Marseille Université (AMU) - Faculté de Médecine, Marseille
Unité VITROME (Vecteurs – Infections Tropicales et Méditerranéennes)
IHU - Méditerranée Infection,
19-21 Boulevard Jean Moulin, 13005 Marseille
Email: florence.fenollar@univ-amu.fr
September 11th, 2023
Dear editor,
Thank you for giving us the opportunity to submit a revised version of our manuscript titled “
Relationship between bacterial vaginosis and sexually trans-mitted infections: coincidence, consequence or co-transmission?”. We are grateful for the time and effort that you and the reviewers have dedicated to providing your valuable feedback on our manuscript. We are indebted to the reviewers for their insightful comments on my review. We have been able to incorporate changes to reflect most of the suggestions provided by the reviewers. We have highlighted the changes within the manuscript. Here is a point-by-point response to the reviewers’ comments and concerns.
Comments for the author:
Reviewer #2:
Authors intended to determine a relation between bacterial vaginosis and STI.
88 references for a paper that is not a revision does not make sense.
We would like to explain that these references include both those of the main article and those of the supplementary data. This approach is in line with the journal's policy, which encourages the inclusion of full references to ensure the transparency and credibility of the research. However, if the journal agrees, we can remove the supplementary data references from the manuscript reference list and place them only in the supplementary data section.
Abstract:
- Authors should not state “There is a correlation”. Instead, they should state something like ‘Our data suggest that a correlation between BV and STI may exist’
As suggested, we replaced " There is a correlation " with "Our data suggest that a correlation between BV and STI may exist".
- “BV itself is not classified as an STI, it can be transmitted during sexual intercourse.” ??? If BV is, as authors state, “a disturbance of the vaginal ecosystem” how can a disturbance be transmitted? This sentence must disappear.
As suggested, we removed this sentence.
Introduction:
- To consider BV as asymptomatic is incongruous. How can one put a name of a disease associated to a laboratory finding without any clinical complaints?
Your point is valid. While BV is often characterized by laboratory findings, it's essential to acknowledge that clinical symptoms or complaints are not always present. BV is indeed a complex condition, and its relationship with symptoms can vary among individuals. Some individuals with BV may experience symptoms, while others may not. The name of the condition is based on microbiological observations, but it's crucial to consider both clinical and microbiological aspects when discussing BV.
- This is surely a problem when talking about BV. Not saying that Nugent or Amsel were better, but the use of molecular techniques to detect and quantify some bacteria (when, to date, the etiology of BV remains mostly unknown) is at least as misleading as older methods… considering some laboratory findings as BV (+) of women with no symptoms or clinical signs and women with symptoms of BV getting a BV (-) result…
You raise a valid concern regarding the diagnosis of BV. BV is indeed a complex condition with diagnostic challenges. While molecular techniques offer a more detailed view of the vaginal microbiota, the interpretation of these findings can be misleading, especially considering that the exact etiology of BV remains largely unknown. The discrepancies you mentioned, where asymptomatic women may test positive for BV, and symptomatic women may test negative, highlight the limitations of current diagnostic methods. This underscores the need for a comprehensive approach that combines clinical evaluation with microbiological data to provide a more accurate diagnosis and improve our understanding of BV.
Materials and Methods:
- When were the vaginal samples taken? During how many days/weeks/months? How long were samples kept before testing? At what temperature?
As suggested, we have added the duration of the collection, when it took place, and the temperature at which we stored them. The new sentence (line 73-74, p 2) reads as follows: “These specimens were retrospectively analysed as part of a six-month data collection period in 2022, and they were stored at -80°C before testing.”
- Age of participants is wrongly in results. And it is repeated in 3.1 and 3.2.
As suggested, we have corrected the age of participants.
- The age is too heterogeneous for this evaluation. Of course, women of more than 70 years old or more might remain sexually active, but women of non-reproductive age (usually above 50) should be excluded, or their evaluation be separated from women 18 to 49. To mix pregnant and non-pregnant also seems incorrect.
As suggested, we removed all women above 50 and we did the analyses separately (pregnant and non-pregnant) and there is a correlation between BV and 4 STIs agents.
- Bacteria and Protozoan names should in italic in lines 105 and 106.
As suggested, we have set in italics.
- To mix HPV among the STI agents is incorrect. Of course it is sexually transmitted but it is not part of the STI screening. How many years can HPV remain in the cervix?... It surely does not correlate with recent infection and this is a reason, among others, for being excluded from the routine STI screening. So, HPV should completely disappear from this paper. The same for Treponema pallidum, Haemophilus ducreyi and HSV1/2 because there is no sense for screening sexually transmitted etiological agents of genital ulcer... without the presence of… genital ulcers… So all the sentences regarding all these agents should disappear
As suggested, we excluded HPV and Mycoplasma hominis from our analysis. Nonetheless, we retained H. ducreyi, T. pallidum, and HSV, even though our patients did not exhibit any ulcer-related symptoms. It's worth noting that, out of curiosity and as part of our routine protocol, we conducted tests for these agents, and we did not detect any cases.
- hominis is part of the vaginal flora… So, why including it as “one microorganism” found? It is normal that it is there. Have authors quantified it?..
As suggested, we removed M. hominis from the manuscript.
- Ln 158 a number with two decimal places
As suggested, we have set the number with two decimal places.
- Figure 1 is of null interest and when HPV disappears, it will no longer make sense
As suggested, we removed Figure 1.
Discussion:
- “Between this dysbiosis with a large panel of microorganisms involved in STIs” large panel?... C. trachomatis, N. gonorrhoeae, T. vaginalis and M. genitalium are 4 and the only that make sense.
As suggested, we change this sentence “a large panel of microorganisms” with “4 microorganisms.”
- Authors cannot state “Our data are thus consistent. ”???. Because authors positive and negative controls?... That is obligatory…
As suggested, we removed this sentence.
- “For STIs, available data indicate that STIs are again on the rise in many EU countries and have been since the mid-1990s [29].” To use a paper published almost 20 years ago… in 2004… for stating that STI are on the rise in 2023… seems very bad. There are much more recent ‘available data’…
As suggested, we added a recent reference: “Geretti, A.M.; Mardh, O.; de Vries, H.J.C.; Winter, A.; McSorley, J.; Seguy, N.; Vuylsteke, B.; Gokengin, D. Sexual transmission of infections across Europe: Appraising the present, scoping the future. Sex. Transm. Infect. 2022, 98, 451–457.”
- The sentence “Finally, one of the strong points of our study is the screening of many different pathogens based on rational molecular tools.” should disappear.
As suggested, we removed this sentence.
- “there is a significant correlation between BV and STIs” when the study will be restricted to 4 STI agents (CT, NG, TV and MG), women 50 and over are discarded, and pregnant discarded or evaluated separately, authors will find out whether there is a correlation or not.
As suggested, we did the analyses separately and there is a correlation between BV and 4 STIs agents.
Sincerely yours,
Pr Florence FENOLLAR
Round 2
Reviewer 1 Report
I still can not see the data about G. vaginalis in results. Please mark it out where it is.
Author Response
September 20th, 2023
Dear editor,
Thank you for giving us the opportunity to submit a revised version of our manuscript titled “
Relationship between bacterial vaginosis and sexually trans-mitted infections: coincidence, consequence or co-transmission?”. We are grateful for the time and effort that you and the reviewers have dedicated to providing your valuable feedback on our manuscript. We are indebted to the reviewers for their insightful comments on my review. We have been able to incorporate changes to reflect most of the suggestions provided by the reviewers. We have highlighted the changes within the manuscript. Here is a point-by-point response to the reviewers’ comments and concerns.
Comments for the author:
Reviewer #1:
- I still can not see the data about G. vaginalis in results. Please mark it out where it is.
As suggested, we added the results of G. vaginalis and F. vaginae and in the results section. The new sentence is as follows at the start of the molecular results section : “BVF was diagnosed in 50 (17.2%) of the 290 women (Table 1). Forty-five of them (90%) had bacterial loads of at least 108 and 109 DNA copies/ml for F. vaginae and G. vaginalis, respectively. Three (6%) had a bacterial load of at least 108 DNA copies/ml for F. vaginae and 2 (4%) of at least 109 DNA copies/ml for G. vaginalis.”
Reviewer 2 Report
Authors solved most of the problems, although 81 references remain and are clearly too much...
Table 3 should only include two sections “pregnant” and “non-pregnant”, the “all” is redundant. If authors want to present some data for “all” women, they can simply describe them in the text.
In table 2 title, authors present their approach correctly “bacterial vaginosis-flora”. I think that the paper would greatly improve if authors created this definition early, as “BVF” in the text and used it whenever they were referring to lab findings. I am aware that the definition of bacterial vaginosis includes the possibility of asymptomatic. But, it is not really reasonable to create a clinical name for something that it is a laboratory finding only. I would greatly appreciate if authors could reflect on this possibility.
Author Response
September 20th, 2023
Dear editor,
Thank you for giving us the opportunity to submit a revised version of our manuscript titled “
Relationship between bacterial vaginosis and sexually trans-mitted infections: coincidence, consequence or co-transmission?”. We are grateful for the time and effort that you and the reviewers have dedicated to providing your valuable feedback on our manuscript. We are indebted to the reviewers for their insightful comments on my review. We have been able to incorporate changes to reflect most of the suggestions provided by the reviewers. We have highlighted the changes within the manuscript. Here is a point-by-point response to the reviewers’ comments and concerns.
Comments for the author:
Reviewer #2:
- Authors solved most of the problems, although 81 references remain and are clearly too much...
As requested, we have lightened the list of references by removing the last references corresponding to those in supplementary table 2. There are now 42 references in the main text, which makes the manuscript clearer. As for the specific references in supplementary table 2, they have been placed just after supplementary table 2, to facilitate reading of the latter and provide the necessary evidence for the various arguments presented in this table.
- Table 3 should only include two sections “pregnant” and “non-pregnant”, the “all” is redundant. If authors want to present some data for “all” women, they can simply describe them in the text.
As suggested, we removed the section “all” in table 3.
- In table 2 title, authors present their approach correctly “bacterial vaginosis-flora”. I think that the paper would greatly improve if authors created this definition early, as “BVF” in the text and used it whenever they were referring to lab findings. I am aware that the definition of bacterial vaginosis includes the possibility of asymptomatic. But, it is not really reasonable to create a clinical name for something that it is a laboratory finding only. I would greatly appreciate if authors could reflect on this possibility.
As suggested, we used the "bacterial vaginosis-flora" approach (BVF) whenever they referred to lab results.
Sincerely yours,
Pr Florence FENOLLAR